# Protective Effects of Jujubosides on 6-OHDA-Induced Neurotoxicity in SH-SY5Y and SK-N-SH Cells

**DOI:** 10.3390/molecules27134106

**Published:** 2022-06-26

**Authors:** Chao-Hsuan Chen, Pei-Chen Hsu, Shih-Wei Hsu, Kun-Ting Hong, Kai-Yuan Chen, Jie-Long He, Der-Yang Cho, Yun-Chi Wang, Wen-Shin Chang, Da-Tian Bau, Chia-Wen Tsai

**Affiliations:** 1Graduate Institute of Biomedical Sciences, China Medical University, Taichung 404333, Taiwan; jemileiia@yahoo.com.tw (C.-H.C.); s1245624@gmail.com (Y.-C.W.); halittlemelon@hotmail.com (W.-S.C.); 2Terry Fox Cancer Research Laboratory, Department of Medical Research, China Medical University Hospital, Taichung 404332, Taiwan; 3Department of Neurosurgery, China Medical University Hospital, Taichung 404333, Taiwan; d5057@mail.cmuh.org.tw; 4Department of Pediatrics, Taoyuan General Hospital, Ministry of Health and Welfare, Taoyuan 33004, Taiwan; peichen@mail.tygh.gov.tw; 5Taichung Armed Forces General Hospital, Taichung 41152, Taiwan; hsushihwei690624@gmail.com; 6Department of Neurological Surgery, Tri-Service General Hospital, Taipei 11490, Taiwan; syndrome1028@gmail.com; 7Department of Neurosurgery, Neurological Institute, Taichung Veterans General Hospital, Taichung 40705, Taiwan; kaiyuan.chen419@gmail.com; 8Department of Post-Baccalaureate Veterinary Medicine, Asia University, Taichung 41354, Taiwan; jielonghe@asia.edu.tw; 9Department of Bioinformatics and Medical Engineering, Asia University, Taichung 41354, Taiwan

**Keywords:** apoptosis, caspase, 6-hydroxydopamine, jujubosides, Parkinson’s disease, reactive oxygen species

## Abstract

6-hydroxydopamine (6-OHDA) is used to induce oxidative damage in neuronal cells, which can serve as an experimental model of Parkinson’s disease (PD). Jujuboside A and B confer free radical scavenging effects but have never been examined for their neuroprotective effects, especially in PD; therefore, in this study, we aimed to investigate the feasibility of jujubosides as protectors of neurons against 6-OHDA and the underlying mechanisms. 6-OHDA-induced neurotoxicity in the human neuronal cell lines SH-SY5Y and SK-N-SH, was used to evaluate the protective effects of jujubosides. These findings indicated that jujuboside A and B were both capable of rescuing the 6-OHDA-induced loss of cell viability, activation of apoptosis, elevation of reactive oxygen species, and downregulation of the expression levels of superoxide dismutase, catalase, and glutathione peroxidase. In addition, jujuboside A and B can reverse a 6-OHDA-elevated Bax/Bcl-2 ratio, downregulate phosphorylated PI3K and AKT, and activate caspase-3, -7, and -9. These findings showed that jujubosides were capable of protecting both SH-SY5Y and SK-N-SH neuronal cells from 6-OHDA-induced toxicity via the rebalancing of the redox system, together with the resetting of the PI3K/AKT apoptotic signaling cascade. In conclusion, jujuboside may be a potential drug for PD prevention.

## 1. Introduction

The etiology of Parkinson’s disease (PD), a neurodegenerative disorder characterized by a progressive loss of dopamine-producing neurons in the substantia nigra of the brain, has not been well studied [1]. It is widely believed that oxidative stress near dopaminergic neurons plays a critical role in the disease [2,3,4,5], and it has been reported that the metabolism of dopamine is the main cause of the imbalance of redox status around the neurons, which at a relatively high level can damage themselves [6,7].

6-hydroxydopamine (6-OHDA), a metabolite of the neurotransmitter dopamine, may cause an accumulation of ROS near the cells themselves and damage dopaminergic neurons [8]. Accumulated ROS include hydrogen peroxide (H_2_O_2_), superoxide radicals (O_2_^•−^) [8,9], and oxidized products [9,10]. In cells and animal models, 6-OHDA has been commonly used as a Parkinson’s experimental model for drug screening and pathway investigation [9,11,12,13]. For instance, 6-OHDA may induce p53- and Bax-mediated apoptotic signaling networks in PC12 cells [14]. In SH-SY5Y cells, 6-OHDA can cause oxidative stress, inducing breakage of the mitochondrial membrane, leakage of cytochrome c, activation of caspase-3, and the promotion of programmed cell death [15]. More interestingly, Cirmi et al. found that 6-OHDA could also induce the production of nitric oxide and affect Parkinson-related genes, such as SNCA, LRRK2, PINK1, DJ-1 and PARK2 [12]. Thus, revealing the underlying mechanisms of 6-OHDA to damage neuronal cells and identifying drugs that can reverse or prevent 6-OHDA-induced damage are important for providing possible strategies against Parkinson’s disease.

Jujubae Fructus (also named jujube, or red date) contains abundant flavonoids, polysaccharides and triterpenic acids. In ancient Chinese culture, jujube has been widely used as a Chinese herbal medicine for thousands of years without scientific evidence. Recent pharmaceutic and pharmacodynamics investigations have shown that flavonoids and polysaccharides are major contributors to the antioxidative efficacy of jujube [16,17,18]. In addition, these polysaccharides in jujube are beneficial to human health because of their immuno-modulatory and hematopoietic efficacies [19,20]. Furthermore, the triterpenic acids present in jujube are the main ingredients responsible for their anti-inflammatory and anticancer properties [21,22]. Betulinic acids and jujuboside Bin jujube have been proven to be beneficial to the cardiovascular system both in cells and animal models [23,24]. In the 2010s, several reports summarized the major components of jujube and their potential health benefits [25,26]. Jujuboside A has been reported to have multiple properties, including antioxidant, anti-inflammatory, anti-anxiety, hypnotic-sedative, and anti-apoptosis capacities [27,28]. Han et al. found that jujuboside A is capable of reducing isoproterenol-induced damage via the PI3K/AKT/mTOR signaling axis [29]. The sedative-hypnotic capacity of jujuboside B is a major characteristic [30,31,32]. In 2014, jujuboside B was first found to have anticancer activity, suppressing the proliferation of AGS and HCT116 cells via the activation of p38 and JNK-mediated apoptotic signals [33]. In 2020, jujuboside B was found to trigger apoptosis in acute leukemia U937 cells via the RIPK1/RIPK3/MLKL signaling pathway [34]. In 2021, Guo et al. reported that jujuboside B can induce apoptosis and autophagy in MDA-MB-231 and MCF-7 human breast cancer cells [35]. Overall, jujubosides A and B, rather than other constituents of jujube, have therapeutic potential in multiple organs and tissues, and translational scientists are just beginning to reveal their mechanisms.

Although several studies have investigated the efficacy of jujuboside A and B in various types of cells, studies focusing on the neuroprotective activities are lacking. In this study, we aimed to reveal the mechanisms by which jujubosides (jujuboside A and B, the main active components in the seeds of wild jujube) prevent the neurotoxic effects of 6-OHDA in SH-SY5Y and SK-N-SH cells.

## 2. Results

### 2.1. 6-OHDA Induced Suppression in Cell Viability

The structures of jujuboside A and B are shown in Figure 1A,B. As shown in Figure 1C,D, treatment with 6.25, 12.5, 25 and 50 μM of 6-OHDA alone for 24 h significantly suppressed SH-SY5Y and SK-N-SH cells’ viability in a dose-dependent manner (Figure 1C,D). In detail, the 6.25, 12.5, 25 and 50 μM 6-OHDA treatments for 24 h were capable of suppressing cell viability to 93.67 ± 1.75, 84.00 ± 4.69, 51.17 ± 8.16, and 16.83 ± 4.26% of untreated SH-SY5Y cells, respectively (Figure 1C). At the same time, 6.25, 12.5, 25, and 50 μM of 6-OHDA treatments for 24 h were capable of suppressing the cell viability to 92.17 ± 5.12, 59.83 ± 9.81, 42.00 ± 9.34 and 11.83 ± 2.64% of untreated SK-N-SH cells, respectively (Figure 1D). 

### 2.2. Alleviation Effects of Jujubosides on 6-OHDA-Induced Decease in Cell Viability

To determine whether jujuboside A and B have protective effects on 6-OHDA-suppressed cell viability, SH-SY5Y and SK-N-SH cells were co-treated with jujuboside A (0, 1, 2, 4, 8, 16, 32, and 64 μM) or jujuboside B (0, 1, 2, 4, 8, 16, 32, and 64 μM) and 25 μM of 6-OHDA for 24 h. As shown in Figure 2, jujuboside A or B alone did not induce a significant suppression of cell viability at a dose of less than 16 μM in either cell line. In addition, they induced less than a 20% loss of cell viability at 32 or 64 μM (Figure 2).

As shown in Figure 3A,B, jujuboside A significantly reversed the cell viability suppressed by 25 μM of 6-OHDA at 4, 8, and 16 μM to 59.83 ± 4.54, 72.67 ± 4.84, and 86.50 ± 3.83% in SH-SY5Y cells, and 60.50 ± 2.66, 73.50 ± 5.13, and 79.83 ± 3.06% in SK-N-SH cells, respectively (Figure 3A,B). Similarly, jujuboside B was capable of rescuing the cell viability suppressed by 25 μM of 6-OHDA at 16, 32 and 64 μM to the levels of 57.83 ± 3.82, 74.17 ± 3.92, and 77.00 ± 5.48% in SH-SY5Y cells, and 59.50 ± 3.45, 75.17 ± 6.40, and 76.50 ± 5.17% in SK-N-SH cells, respectively (Figure 3C,D).

### 2.3. 6-OHDA Would Induce Cell Apoptosis

To examine whether 6-OHDA-suppressed cell viability was induced by the induction of apoptosis, the appearance of sub-G1 cells was measured by flow cytometry. The results showed that 6-OHDA could induce cells to undergo apoptosis at the dose above 12.5 μM and 6.25 μM in SH-SY5Y and SK-N-SH cells, respectively (Figure 4). In detail, 6-OHDA 12.5, 25, and 50 μM treatments were capable of inducing 14.52 ± 6.93, 38.75 ± 5.77, and 72.75 ± 12.74% apoptosis of SH-SY5Y cells, respectively. At the same time, 6-OHDA 12.5, 25, and 50 μM treatments were capable of inducing 19.55 ± 6.39, 44.62 ± 5.95, and 73.92 ± 7.92% apoptosis of SK-N-SH cells, respectively.

### 2.4. Rescuing Effects of Jujubosides on 6-OHDA-Induced Cell Apoptosis

We selected the most obvious doses of 6-OHDA (25 and 50 μM) to induce sub-G1 cells (Figure 4) for further investigation. In Figure 5, it was observed that co-treatment with 4, 8 and 16 μM of jujuboside A and 16, 32 and 64 μM of jujuboside B can effectively suppress the 6-OHDA-induced appearance of sub-G1 apoptotic SH-SY5Y, and SK-N-SH cells (Figure 5). In detail, 4, 8 and 16 μM of jujuboside A treatments were capable to reverse 50 μM 6-OHDA-induced apoptotic cells to the level of 54.72 ± 2.90, 35.53 ± 5.69 and 26.12 ± 4.01% of intact SH-SY5Y cells, respectively (Figure 5A). The same doses of 4, 8, and 16 μM of jujuboside A treatments were capable to reverse 50 μM 6-OHDA-induced apoptotic cells to the level of 64.42 ± 4.60, 36.92 ± 6.63 and 23.43 ± 3.81% of intact SK-N-SH cells, respectively (Figure 5B). At the same time, 16, 32, and 64 μM of jujuboside B treatments were capable to reverse 50 μM 6-OHDA-induced apoptotic cells to the level of 65.20 ± 4.22, 60.92 ± 5.86, and 50.17 ± 7.09% of intact SH-SY5Y cells, respectively (Figure 5C). The same doses of 16, 32 and 64 μM of jujuboside B treatments were capable to reverse 50 μM 6-OHDA-induced apoptotic cells to the level of 66.13 ± 7.97, 63.58 ± 7.07 and 56.93 ± 4.81% of intact SK-N-SH cells, respectively (Figure 5D). Similar patterns of alterations were observed for 25 μM 6-OHDA-induced apoptosis in both the SH-SY5Y and SK-N-SH cells (Figure 5).

### 2.5. 6-OHDA Induced Intracellular ROS Elevation

To investigate the time-dependent elevation of 6-OHDA-induced ROS in SH-SY5Y, and SK-N-SH cells, the cells were treated with 10 and 25 μM of 6-OHDA for 0.5, 1, 2, 4, 12, and 24 h, and intracellular ROS levels were measured at the indicated time points. Evidently, 25 μM of 6-OHDA induced an increase in ROS in both the SH-SY5Y and SK-N-SH cells compared to 10 μM of 6-OHDA. The data showed that the elevation of ROS was dose dependent. In addition, the ROS peaks appeared at 12 h and lasted for 24 h in both cell types, and the SK-N-SH cells produced more ROS than the SH-SY5Y cells (Figure 6).

### 2.6. Eliminative Effects of Jujubosides on 6-OHDA-Induced ROS

To determine whether jujuboside A and B have eliminative effects on 6-OHDA-induced ROS production, SH-SY5Y and SK-N-SH cells were co-treated with jujuboside A (0–16 μM) or jujuboside B (0–64 μM) and 25 μM of 6-OHDA for 24 h. As shown in Figure 7, jujuboside A or B alone did not induce any elevation or suppression of ROS production at any dose tested in either cell lines (Figure 7).

As shown in Figure 8A,B, jujuboside A significantly reduced the ROS levels elevated by 25 μM of 6-OHDA at 4, 8, and 16 μM in both the SH-SY5Y and SK-N-SH cells (Figure 8A,B). Similarly, jujuboside B reduced the ROS produced by 25 μM of 6-OHDA at 16, 32, and 64 μM in both the SH-SY5Y and SK-N-SH cells (Figure 8C,D).

### 2.7. Alterations of Apoptotic-Related and Redox-Related Proteins

The expression levels of apoptosis-related and redox-related proteins were detected by Western blotting to reveal the signaling network induced by the treatment with 6-OHDA and jujubosides. First, phosphorylated PI3K and AKT were suppressed by a 25 μM 6-OHDA 24 h treatment, and rebounded by co-treatment with jujubosides in SH-SY5Y cells (Figure 9). Second, superoxide dismutase (SOD), catalase (CAT) and glutathione peroxidase (GPx) were also suppressed by a 25 μM 6-OHDA 24 h treatment, and rebounded by co-treatment with jujubosides (A at 16 μM and B at 64 μM) in SH-SY5Y cells (Figure 10). Third, the pro-apoptotic protein Bax was upregulated by a 6-OHDA 24 h treatment, and suppressed by co-treatment with jujubosides in SH-SY5Y cells (Figure 11). At the same time, the anti-apoptotic protein Bcl2 seemed to be suppressed by the 6-OHDA 24 h treatment and rebounded by co-treatment with jujubosides in SH-SY5Y cells (Figure 11). Lastly, the active forms of caspase-3, -8, and -9 were obviously overexpressed by the 6-OHDA 24 h treatment, and suppressed by co-treatment with jujubosides in SH-SY5Y cells (Figure 12). The changes in these proteins measured in the SH-SY5Y cells were the same as those in the SK-N-SH cells (data not shown).

## 3. Discussion

To the best of our knowledge, the current study is the first to reveal the neuroprotective effects and mechanisms of jujubosides in 6-OHDA-challenged SH-SY5Y and SK-N-SH cells. SH-SY5Y is a widely used cell line that mimics PD at the cellular level [36]. The SK-N-SH cell line is the parental cell line from which SH-SY5Y cells are sub-cloned and was originally established from a bone marrow biopsy of a neuroblastoma case [37]. Both cell lines are not well differentiated cells; however, they are useful in neuroscience because it is not easy to keep neural cells proliferating or even alive.

In the 6-OHDA-induced PD cellular model, we systematically examined the feasibility of applying jujubosides to protect neuronal cells from the attack of 6-OHDA in multiple aspects. First, we revealed that both jujubosides A and B protected SH-SY5Y and SK-N-SH cells from 6-OHDA-suppressed cell viability (Figure 3). In addition, jujubosides A and B reversed the 6-OHDA-induced apoptosis (Figure 5). As for the 6-OHDA-induced ROS, jujuboside A and B effectively reversed those ROS elevated by 6-OHDA in the SH-SY5Y and SK-N-SH cells (Figure 8). As for the detailed mechanisms, 6-OHDA-suppressed phosphorylation of PI3K, and AKT was rebounded when the jujubosides were co-treated with 6-OHDA (Figure 9A). These antioxidant enzymes, including SOD, CAT and GPx, were downregulated by the 6-OHDA and their expression was restored by co-treatment with jujubosides and 6-OHDA (Figure 9B). The elevated Bax/Bcl2 ratio induced by the 6-OHDA was reversed by co-treatment with jujubosides (Figure 9C).

Redox imbalances may be the main toxicity caused by 6-OHDA in neuronal cells. In our study, treatments of 25 and 50 μM of 6-OHDA to both SH-SY5Y and SK-N-SH cells elevated dose-dependent intracellular ROS, which was reversed by co-treatment with jujubosides (Figure 6 and Figure 8). Jujubosides did not cause any obvious changes in the SH-SY5Y and SK-N-SH cells (Figure 7). Although we have no direct evidence of the source of the ROS, it is hypothesized that the elevation of ROS is a direct result of mitochondrial impairment [38]; however, there are still other possible mechanisms, such as intracellular enzymes (e.g., NADPH oxidases), the electron transportation chain reaction in mitochondria and hydrogen peroxide metabolism, which require further validation. Simultaneously, we found that antioxidant enzymes, including SOD, CAT, and GPx, were downregulated by 6-OHDA treatment and their expression was restored by co-treatment with jujubosides and 6-OHDA (Figure 9B). The downregulation of SOD and CAT by 6-OHDA in the SH-SY5Y cells is consistent with that reported by Crimi [12]. The alterations of GPx in the SH-SY5Y and SK-N-SH cells strengthened the concept that GPx is also involved in 6-OHDA neurotoxicity, and redox impairment may be critical and does not have cell specificity. The superoxide radical, a type of ROS, can interact with nitric oxide to form a more toxic molecule, peroxynitrite, to attack neuronal cells non-specifically [39]. It has been reported that 6-OHDA can also produce nitric oxide in SH-SY5Y cells [12]. The role of peroxynitrite in 6-OHDA-induced neurotoxicity and PD etiology requires further investigation.

The 6-OHDA-induced neuro-damaging signaling network of apoptosis is complex and interesting. According to the KEGG prediction for 6-OHDA-induced signals, the outcomes indicated that the top signaling pathway cluster is PI3K/AKT, which is involved in cell growth, survival, and proliferation [40]. We hypothesized that the PI3K/AKT signaling pathway is involved in the neuroprotective effects of 6-OHDA-induced toxicity. The results showed that 6-OHDA significantly suppressed the expression of phosphorylated PI3K and AKT, which could be rescued by the addition of jujubosides together with 6-OHDA (Figure 9A). An elevated Bax/Bcl2 ratio was found (Figure 9C), although the changes in Bcl2 by the 6-OHDA were not as obvious as Chen’s [40], which is consistent with our previous report [13]. This significant increase in Bax is consistent with the results of Chen et al. [13,40]. Activated caspase-3, -7, and -9 were observed and downregulated to basal levels by co-treatments of jujuboside (Figure 12), whereas alterations in caspase-1, -2 and, -8 were not obvious (data not shown). Notably, the elevated Bax/Bcl2 ratio and activated caspase-3, -7, and -9 were reversed by a co-treatment with jujuboside A or B. The overall network of how 6-OHDA induced neurotoxicity and how jujubosides rescue these effects is summarized in Figure 13.

In 2018, Wan et al. examined the effects of jujuboside A on norepinephrine-induced loss of cell viability and apoptosis in rat H9c2 cardiomyocytes. The cells were pretreated with jujuboside A and it was found that jujuboside A was capable of reversing norepinephrine-induced loss of cell viability and apoptosis. This is the first study to provide solid evidence for jujuboside A as a potential therapeutic strategy for the treatment of heart disease. Jujuboside A altered the signaling molecules, including p-AKT, p-ERK, p-p38, p-c-Jun, Bax, Bcl-2 and cleaved caspase-3 and -9 [41]. In the current study, we had a higher clinical practicability than their methodology in the following aspects: (a) both jujuboside A and B were examined and proved to be feasible; (b) in human SH-SY5Y and SK-N-SH neuronal cells rather than rat cells; (c) we applied the co-treatment protocol rather than a pre-treatment. The norepinephrine-induced alterations and 6-OHDA-induced signaling may share some common pathways, such as p-AKT, Bax, Bcl-2, cleaved caspase-3 and -9. We examined the involvement of the antioxidant enzymes CAT, SOD, and GPx and found the involvement of cleaved caspase-7 and p-PI3K.

Permeability of the blood–brain barrier is frequently the most important issue in novel drug discovery. We are also interested in the feasibility of jujubosides to penetrate the blood–brain barrier. Currently, there is a lack of evidence in animal models to test whether jujubosides can cross the blood–brain barrier via any mechanism; however, several lines of evidence show that jujuboside A can influence the brain area. For instance, as early as 2002, jujuboside A was recognized for its inhibitory effect on paired-pulse responses of dentate gyrus granule cells in the hippocampus of rats [42]. In 2019, jujuboside A treatment effectively prevented memory impairment in a mouse model [43]. Daily treatment with 300 mg/kg of jujuboside A has been proven to effectively ameliorate high-fat-diet and streptozotocin-induced diabetic nephropathy in SD rats [44]. The structure of jujuboside B is similar to that of jujuboside A (Figure 1A,B). Thus, it is possible that jujuboside B can cross the blood–brain barrier through mechanisms similar to those of jujuboside A.

In summary, this study is the first to demonstrate that jujubosides exert neuroprotective effects by suppressing the apoptosis of neuronal cells induced by 6-OHDA. In detail, we have shown that 6-OHDA-induced redox imbalance together with PI3K/AKT/caspase-related apoptotic signaling can be rescued by both jujuboside A and B. More investigations are needed to validate the specific intracellular mechanisms underlying the protective effects of the jujubosides in PD. From a therapeutic viewpoint, jujuboside supplementation could be a potential non-toxic strategy for the treatment of PD.

## 4. Materials and Methods

### 4.1. Cell Culturing Conditions

The SH-SY5Y cells were purchased from American Type Culture Collection (ATCC, Rockville, MD, USA) and cultured at 37 °C in minimum essential media (MEM) supplemented with 10% heat-inactivated fetal bovine serum (FBS, Mediatech Inc., Herndon, VA, USA) [13]. The SK-N-SH cells were also purchased from the American Type Culture Collection and cultured in Dulbecco’s Modified Eagle/F12 medium (Sigma Chemical Co., St Louis, MO, USA) as previously described [45].

### 4.2. Chemicals and Common Treatment Protocols

Chemicals, including the 6-OHDA, jujuboside A, jujuboside B, dimethyl sulfoxide (DMSO), propidium iodide (PI) and 3-(4,5-dimethylthiazol-2-yl)-2,5-diphenyl-tetrazolium bromide (MTT), were obtained from Sigma Chemical Co. (St. Louis, MO, USA) and Aldrich Chemical Co. (Milwaukee, WI, USA). The 6-OHDA (10 mM stock solution) was diluted in the medium to obtain concentrations ranging from 10 to 100 μM [13]. The jujubosides were freshly dissolved in DMSO, and further diluted to several dosages with the medium for the cell treatment.

### 4.3. Determination of Cell Viability by MTT Assay

Cell viability was assessed using a tetrazolium 3-(4,5-dimethylthiazole-2-yl)- 2,5-diphenyltetrazolium bromide (MTT) assay as previously described [46,47]. Briefly, the cells were cultured in 96-well plates at a density of 3 × 10^4^ cells/well, grown for another day and then exposed to 25 or 50 μM of 6-OHDA, with or without treatment with jujubosides at different concentrations for 1 h prior to 6-OHDA exposure. Twenty-four hours after the 6-OHDA exposure, the medium was removed and replaced with a final concentration of 0.5 mg/mL of MTT. The plates were incubated for 4 h in a humidified atmosphere at 37 °C, and 5% CO_2_. The color intensity was measured at 570 nm using a Multiskan MS ELISA reader (Labsystems, Helsinki, Finland).

### 4.4. Measurement of ROS Production

The SH-SY5Y and SK-N-SH cells were plated at a density of 2 × 10^5^ cells/well into 12-well plates and incubated with 6-OHDA alone or in combination with jujubosides for 24 h. The cells were then harvested and resuspended in 500 μL DCFH-DA (10 μM), incubated at 37 °C for 30 min, and analyzed by flow cytometry to detect the intracellular ROS, as previously described [45].

### 4.5. Determination of Apoptosis

In total, 2 × 10^6^/mL SH-SY5Y and SK-N-SH cells were seeded in 10 cm dishes with 6-OHDA alone, or with a jujuboside co-treatment. After 24 h, the cells were harvested, fixed gently with 70% ethanol, incubated with a PI buffer (4 μg/mL PI, 0.5 μg/mL RNase, and 1% Triton X-100 in PBS), filtered through a 40 μm nylon filter, and 10,000 PI-stained cells in each experiment were detected for the appearance of the sub-G1 phase using a FACS Calibur instrument (BD Biosciences, San Jose, CA, USA) equipped with Cell Quest software as described previously [48,49].

### 4.6. Western Blotting Procedure

A Bio-Rad protein assay system (Bio-Rad Laboratories, Inc.) was used to determine the protein concentrations. The samples (35 μg per lane) were separated by 10–12% SDS-PAGE and transferred to a polyvinylidene fluoride (PVDF) membrane using a transfer apparatus. The bands were cut into streams by molecular weight, according to the rainbow markers. The protocols for blocking and incubating with primary and secondary antibodies were the same as previously published [50,51,52]. Primary antibodies including phospho-AKT(S473) (sc-293125, 1:1000), Bcl2 (sc-7382, 1:2000), Bax (sc-7480, 1:500), SOD (sc-101523, 1:1000), CAT (sc-271358, 1:2000), GPx (sc-133160, 1:1000), cleaved caspase-3 (sc-56052, 1:1000), cleaved caspase-9 (sc-56073, 1:2000), β-actin (sc-47778, 1:1000), and horseradish peroxidase (HRP)-conjugated secondary antibodies (sc-2031, 1:15000) were purchased from Santa Cruz Biotechnology (Santa Cruz, CA, USA). Antibodies against phospho-PI3K (4228, 1:2000), and cleaved caspase-7 (9491, 1:2000) were purchased from Cell Signaling Technology, Inc. (Danvers, MA, USA). Finally, the bands were visualized with enhanced chemiluminescence (ECL) and the ImageJ 1.52v program was used for the quantitative analysis of the intensity of the band signal.

### 4.7. Statistical Methodology

Statistical significance was assessed using a *Student’s*
*t*-test and one-way ANOVA followed by a *post hoc* test. The results are plotted as the mean ± SEM and any value of *p* < 0.05 was considered statistically significant and is shown with a star mark in the Figures.

## Figures and Tables

**Figure 1 molecules-27-04106-f001:**
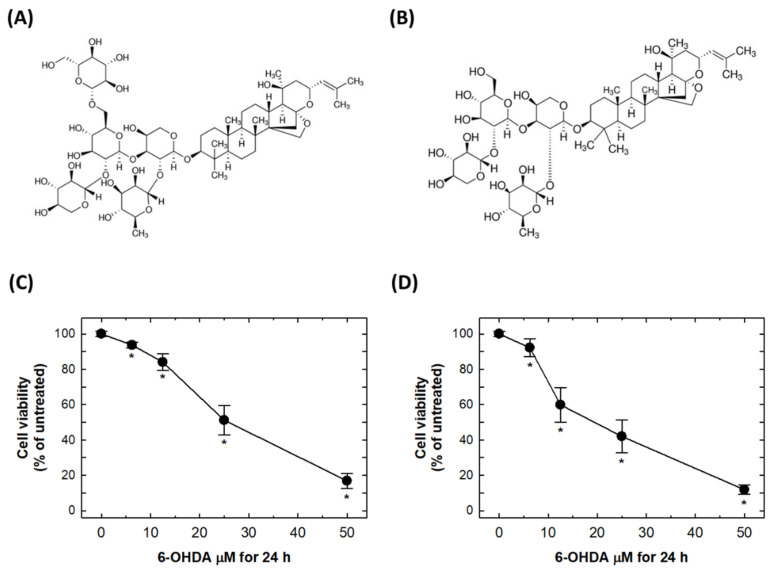
Structures of jujuboside A and B, and 6-OHDA suppressive effects on cell viabilities in SH-SY5Y, and SK-N-SH cells. (**A**,**B**) Structures of jujuboside A and B, respectively. (**C**,**D**) 6-OHDA suppressive effects on cell viabilities in SH-SY5Y and SK-N-SH cells, respectively. * Statistically significant from untreated group. Each experiment was repeated for at least 6 times.

**Figure 2 molecules-27-04106-f002:**
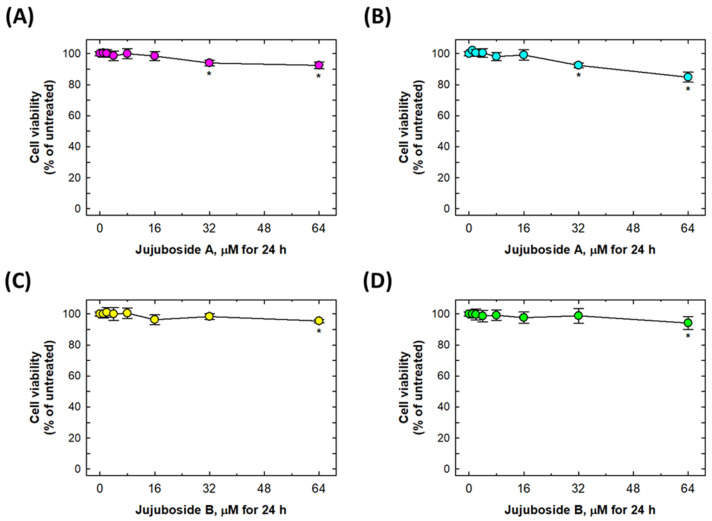
Suppressive effects of jujuboside A and B on cell viabilities in SH-SY5Y and SK-N-SH cells. (**A**,**B**) Suppressive effects of jujuboside A on SH-SY5Y and SK-N-SH cells, respectively. (**C**,**D**) Suppressive effects of jujuboside B on SH-SY5Y and SK-N-SH cells, respectively. * Statistically significant from untreated group. Each experiment was repeated for at least 6 times.

**Figure 3 molecules-27-04106-f003:**
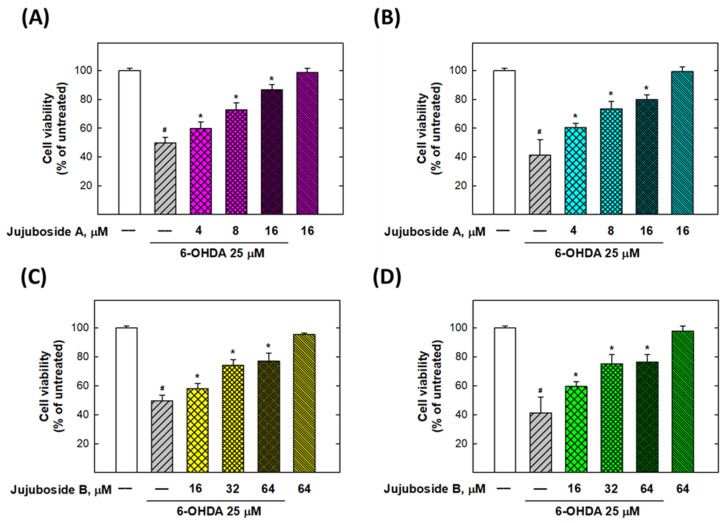
Jujuboside A and B reversed the 6-OHDA-induced loss of cell viability in SH-SY5Y and SK-N-SH cells. (**A**,**B**) Effects of jujuboside A on SH-SY5Y and SK-N-SH cell viabilities, respectively. (**C**,**D**) Effects of jujuboside B on SH-SY5Y and SK-N-SH cell viabilities, respectively. * Statistically significant from 6-OHDA 25 μM group. # Statistically significant from untreated group. Each experiment was repeated for at least 6 times.

**Figure 4 molecules-27-04106-f004:**
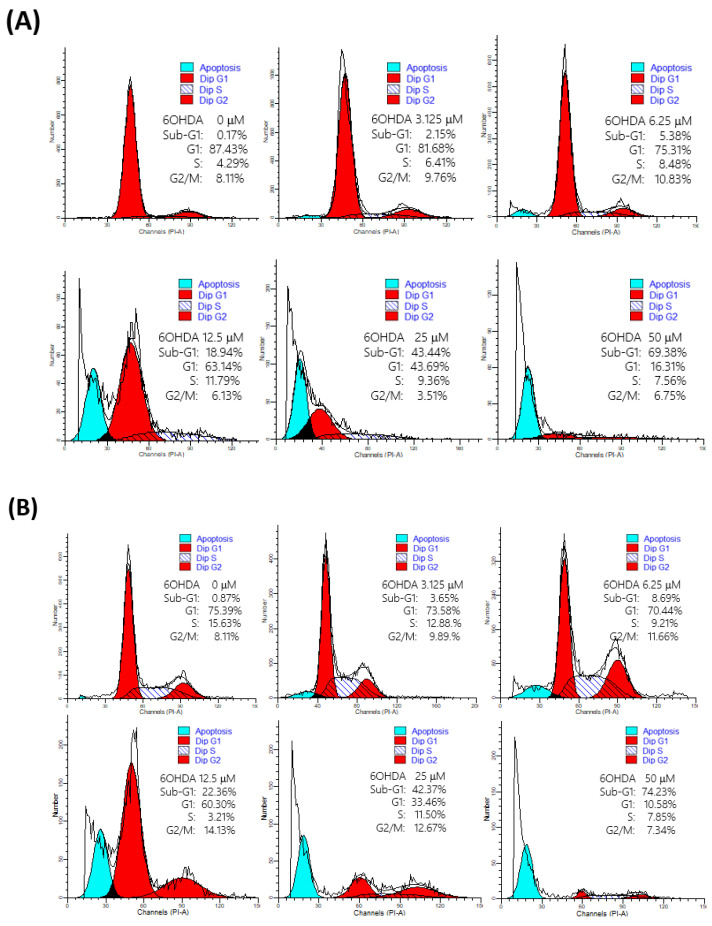
6-OHDA dose-dependently induced apoptosis in SH-SY5Y and SK-N-SH cells. (**A**) After treatment with 6-OHDA at the indicated concentrations, SH-SY5Y cells were harvested, stained with propidium iodide (PI), and analyzed by flow cytometry. (**B**) After treatment with 6-OHDA at the indicated concentrations, SK-N-SH cells were harvested, stained with PI, and analyzed by flow cytometry. (**C**) The effect of 6-OHDA on SH-SY5Y cell apoptosis. (**B**,**D**) Effects of 6-OHDA on SK-N-SH cell apoptosis. * Statistically significant compared with the untreated group. Each experiment was repeated at least six times.

**Figure 5 molecules-27-04106-f005:**
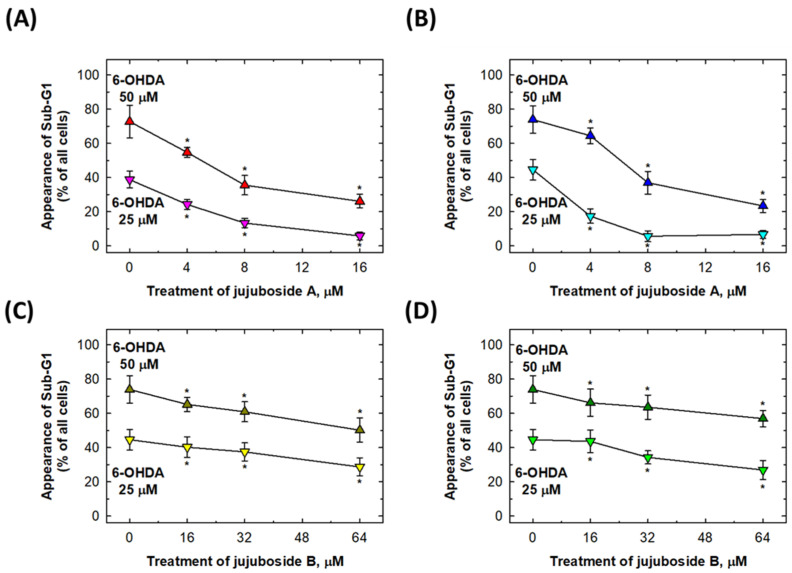
Jujuboside A and B reversed the 6-OHDA-induced cell apoptosis in SH-SY5Y and SK-N-SH cells. (**A**,**B**) Effects of jujuboside A on SH-SY5Y and SK-N-SH cell apoptosis, respectively. (**C**,**D**) Effects of jujuboside B on SH-SY5Y and SK-N-SH cell apoptosis, respectively. * Statistically significant from 6-OHDA group. Each experiment was repeated for at least 6 times.

**Figure 6 molecules-27-04106-f006:**
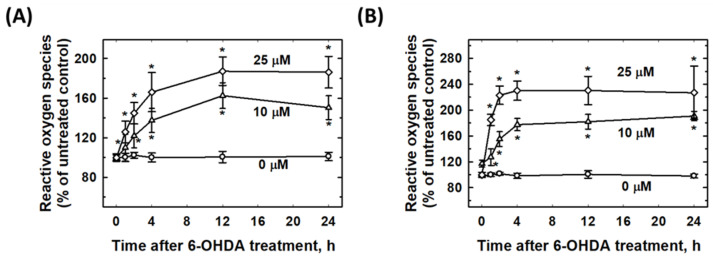
6-OHDA induced the elevation of reactive oxygen species (ROS) in dose-dependent manner in SH-SY5Y and SK-N-SH cells. (**A**) Induction of 6-OHDA expression in SH-SY5Y intracellular ROS. (**B**) Induction of 6-OHDA on SK-N-SH intracellular ROS. * Statistically significant compared with the untreated group. Each experiment was repeated at least six times.

**Figure 7 molecules-27-04106-f007:**
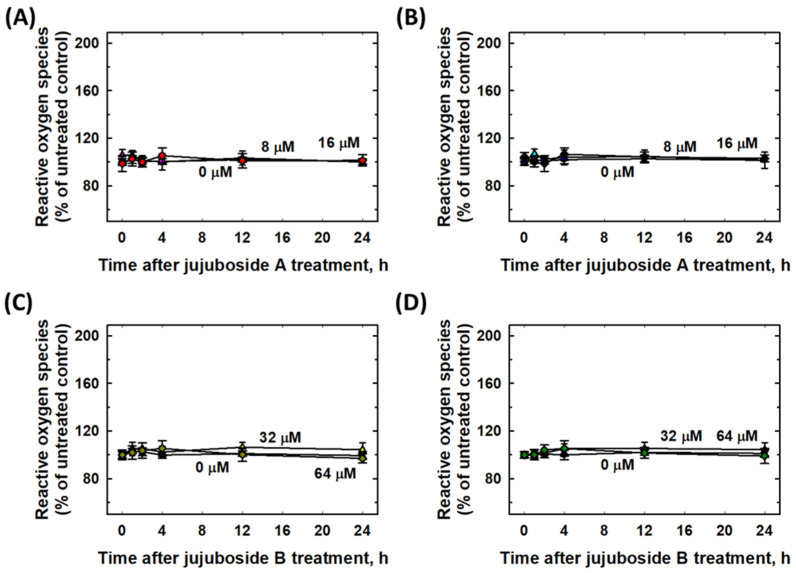
Effects of jujuboside A and B on intracellular reactive oxygen species (ROS) in SH-SY5Y and SK-N-SH cells. (**A**,**B**) Effects of jujuboside A on intracellular ROS of SH-SY5Y and SK-N-SH cells, respectively. (**C**,**D**) Effects of jujuboside B on intracellular ROS of SH-SY5Y and SK-N-SH cells, respectively. Each experiment was repeated for at least 6 times.

**Figure 8 molecules-27-04106-f008:**
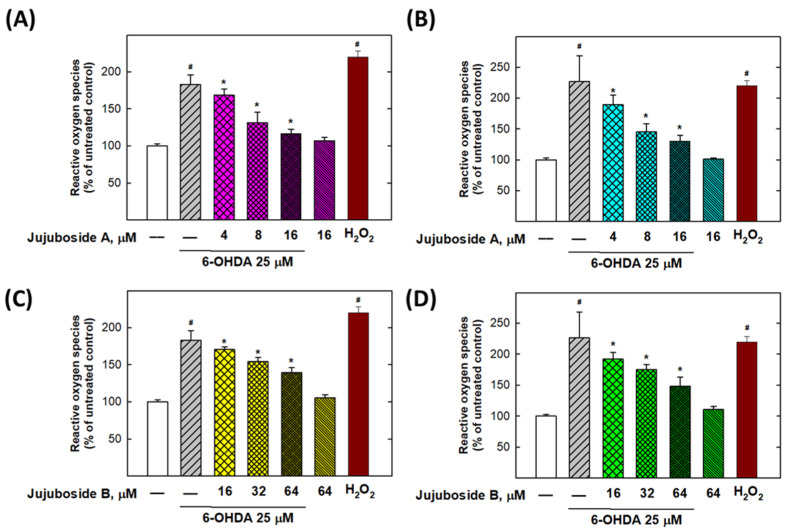
Jujuboside A and B reversed the 6-OHDA-induced reactive oxygen species (ROS) in SH-SY5Y and SK-N-SH cells. (**A**,**B**) Effects of jujuboside A on SH-SY5Y and SK-N-SH intracellular ROS, respectively. (**C**,**D**) Effects of jujuboside B on SH-SY5Y and SK-N-SH intracellular ROS, respectively. * Statistically significant from 6-OHDA 25 μM group. # Statistically significant from untreated group. Each experiment was repeated for at least 6 times.

**Figure 9 molecules-27-04106-f009:**
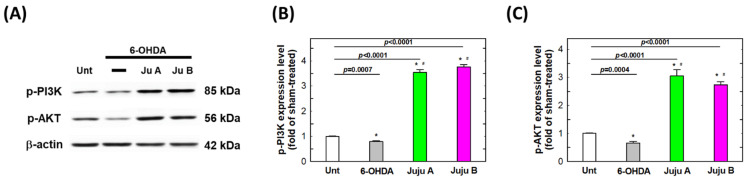
The signaling patterns altered by the treatments of 6-OHDA and jujubosides. (**A**) Western blotting was conducted specifically for p-PI3K and p-AKT. (**B**) Quantitative analysis of the intensities of p-PI3K bands. (**C**) Quantitative analysis of the intensities of p-AKT bands. β-actin served as an internal loading control in each repeat of experiment. * significant different from untreated group. # significant different from the 6-OHDA treated group.

**Figure 10 molecules-27-04106-f010:**
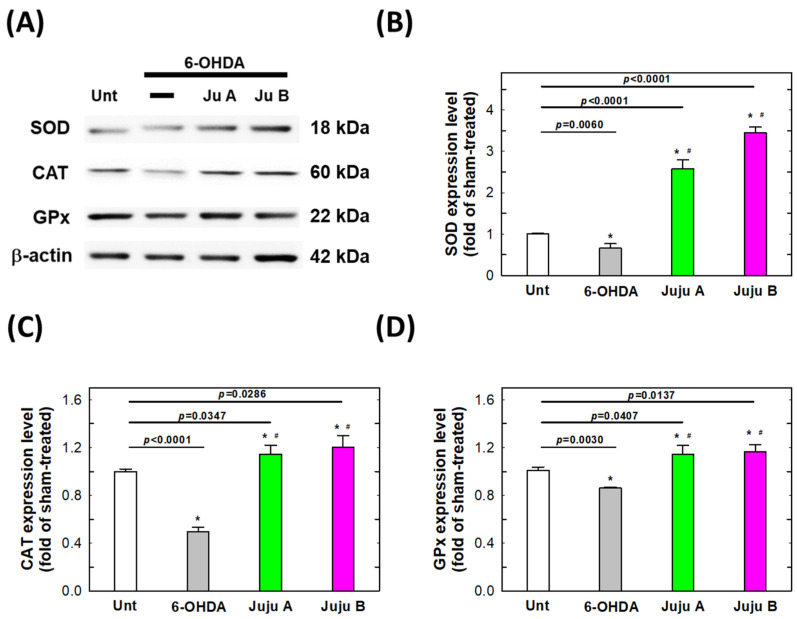
The signaling patterns altered by the treatments of 6-OHDA and jujubosides. (**A**) Western blotting was conducted specifically for SOD, CAT and GPx. (**B**) Quantitative analysis of the intensities of SOD bands. (**C**) Quantitative analysis of the intensities of CAT bands. (**D**) Quantitative analysis of the intensities of GPx bands. β-actin served as an internal loading control in each repeat of experiment. * significant different from untreated group. # significant different from the 6-OHDA treated group.

**Figure 11 molecules-27-04106-f011:**
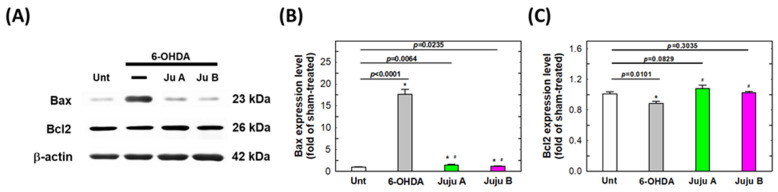
The signaling patterns altered by the treatments of 6-OHDA and jujubosides. (**A**) Western blotting was conducted specifically for Bax and Bcl2. (**B**) Quantitative analysis of the intensities of Bax bands. (**C**) Quantitative analysis of the intensities of Bcl2 bands. β-actin served as an internal loading control in each repeat of experiment. * significant different from untreated group. # significant different from the 6-OHDA treated group.

**Figure 12 molecules-27-04106-f012:**
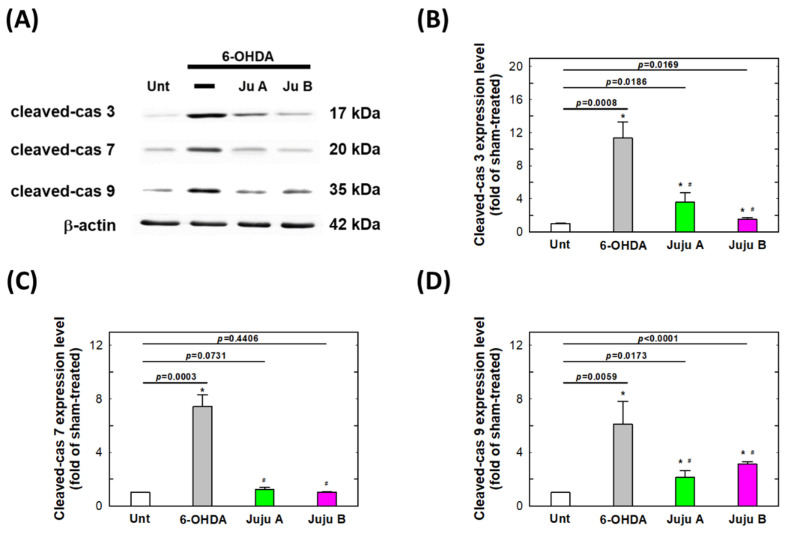
The signaling patterns altered by the treatments of 6-OHDA and jujubosides. (**A**) Western blotting was conducted specifically for cleaved caspase-3, cleaved caspase-7 and cleaved caspase-9. (**B**) Quantitative analysis of the intensities of cleaved-caspase 3 bands. (**C**) Quantitative analysis of the intensities of cleaved-caspase 7 bands. (**D**) Quantitative analysis of the intensities of cleaved-caspase 9 bands. β-actin was served as an internal loading control in each repeat of experiment. * significant different from untreated group. # significant different from the 6-OHDA treated group.

**Figure 13 molecules-27-04106-f013:**
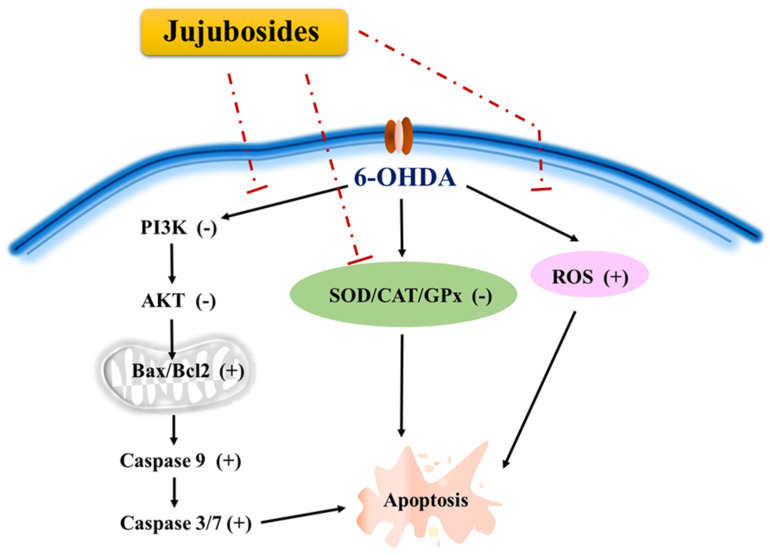
The overall network of how the 6-OHDA induced neurotoxicity and how jujubosides conducted their rescuing effects.

## Data Availability

The data presented in this study are available upon request from the corresponding author.

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
