# Peer review of "Protective Effects of Jujubosides on 6-OHDA-Induced Neurotoxicity in SH-SY5Y and SK-N-SH Cells"

_molecules, 2022, doi:10.3390/molecules27134106_

Round 1

Reviewer 1 Report

This manuscript by Chen and colleagues describes how jujubosides A and B can prevent the neurotoxic effects of 6-OHDA in a cultured neuron model. The study is interesting and, for the most part, well reported. My main comments relate to western blotting in the study, which is not conducted or reported as rigorously as other experiments.

Major comments

·         Western blotting:

o   The western blotting data presented are not quantitative and do not support the conclusions drawn. To report up- or down-regulation of proteins by 6-OHDA and their normalisation by jujubosides, the authors must include sufficient replicates, quantify the optical density of bands and analyse these data statistically. Even though western blots are at best semi-quantitative, quantitation must be conducted to support the authors’ conclusions.

o   Please report your western blot data by showing the entirety of some representative lanes, alongside the molecular weight marker used, in the paper. The figure provided contains images which have been heavily modified, and it does not accurately represent the western blotting conducted.

o   Please provide original blot images for reviewers. The ‘original images for blots/gels’ file provided for reviewers does NOT contain original images but only contains the heavily modified images already provided in the manuscript.

o   In the methods, please describe the western blotting protocol in more detail and state the specific primary and secondary antibodies used (supplier, cat #) and their concentrations rather than simply referring to other papers.

·         There seems to be scope to describe relevant literature in more detail. Some findings replicate work by Wan et al 2018 in cardiomyocytes (https://doi.org/10.3892/mmr.2017.7938) which is not cited.

Minor comments

·         In the introduction (p2 line 67 onwards) it would be helpful to describe what jujuboside A and B are and explain why these, rather than other constituents of jujube, are of therapeutic interest

·         Please indicate the p value for comparisons shown in figures

·         The paper could benefit from review of English language, eg-

o   the use of prepositions. For example, in the abstract, “Jujuboside A and B 26 confer the free radical scavenging effects while never have been checked for its neuroprotective 27 effects, especially that for PD” should read “Jujuboside A and B 26 confer free radical scavenging effects but have never have been checked for their neuroprotective 27 effects, especially in PD

o   the first sentence of the introduction contains no verb in the main clause

o   other issues throughout manuscript

·         Typographic errors, the ‘micro’ symbol is missing from p2 line 98 and p9 line 236

Reviewer 2 Report

The article is well structured and documented, being in line with the guidelines for the authors, imposed by the journal. The abstract is well structured and clearly presented, according to this research. The state-of-the-art presented in Introduction of the paper is well documented and focused on  the actual research direction in the field. The experimental procedure is well justified and comprehensibly showed. The results are presented in a concise manner and sustained by proper figures and tables. Furthermore, the statistical analysis indicates that the results obtained are statistically significant. The discussion section is clear, understandable and in accordance with the results obtained. The conclusion is well explained by the results of the experiments performed. The references indicate a very well documentation.

Reviewer 3 Report

Review of the paper molecules-1765488

This manuscript reported the protective effects of jujubosides A and B on 6-OHDA-induced neurotoxicity in SH-SY5Y and SK-N-SH cells, especially that for Parkinson’s disease. Even the topic is interesting, the reviewer has some questions / comments concerning this work:

-In 2.4. “Rescuing Effects of Jujubosides on 6-OHDA-induced Cell Apoptosis”, the stained cells were analyzed using flow cytometer and the original data should be presented.

-The concentration unit μM was missing in line 98, 115, 117, etc.

- The coordinates in Figs. 1-8 should be rectified according to the literature.

- In Fig. 9, internal standard β-actin were quite different from each other.

-The resolution of figures should be enhanced. e.g. Fig. 1 AB

Round 2

Reviewer 1 Report

Thank you for responding to my feedback. I still have concerns about the Western blotting in the study.

Please clarify:

- What was the nature of replication in this part of the study? Were independent replicate experiments conducted and if so, how many? Were protein extracts from each replicate run in separate gel lanes then transferred, imaged and quantified separately? Are these represented in the error bars in figures 9-12? The error bars appear unrealistically small and in fig 12 are non-existent, calling into question how replication was conducted and whether independent replicate experiments were used. If independent replicate experiments were used (as they should be) then raw images of these replicates should be provided for reviewers.

- Were bands normalised numerically to the intensity of beta actin on the same blot? There is no mention of this necessary normalisation in the methods. By eye at least, the expression levels in fig 11C appear to correspond to the Bcl2 band intensity in fig 11A. However, the actin intensity in figure 11 varies between lanes, indicating different loading amounts of protein. If this were used for normalisation, Bcl2 expression levels may not appear as shown in fig 11C.

- how were SOD and GPX probed for on the same membrane without stripping (fig 10)? They are presented with the same beta actin control suggesting they were probed on the same blot. But from my experience their molecular weight difference (~4kDa) would  be almost impossible to resolve on a 10-12% SDS PAGE gel, which would mean it would be impossible to guess where to cut the blot to.

- similarly, how were Bax and Bcl2 (fig 11, ~3kDa different) and cas3 and cas7 (fig 12, ~3kDa different) probed for on the same blots?